# AN EFFICIENT IMPLEMENTATION FOR SOLVING THE ALL PAIRS MINIMAX PATH PROBLEM IN AN UNDIRECTED DENSE GRAPH

## ABSTRACT

We provide an efficient $O(n^2)$ implementation for solving the all pairs minimax path problem or widest path problem in an undirected dense graph. It is a code implementation of the Algorithm 4 (MMJ distance by Calculation and Copy) in a previous paper. The distance matrix is also called the all points path distance (APPD). We conducted experiments to test the implementation and algorithm, compared it with several other algorithms for solving the APPD matrix. Result shows Algorithm 4 works good for solving the widest path or minimax path APPD matrix. It can drastically improve the efficiency for computing the APPD matrix. There are several theoretical outcomes which claim the APPD matrix can be solved accurately in $O(n^2)$. However, they are impractical because there is no code implementation of these algorithms. It seems Algorithm 4 is the first algorithm that has an actual code implementation for solving the APPD matrix of minimax path or widest path problem in $O(n^2)$, in an undirected dense graph.

## 1 INTRODUCTION

The minimax path problem is a classic problem in graph theory and optimization. It involves finding a path between two nodes in a weighted graph such that the maximum weight of the edges in the path is minimized. [1]

Given a graph $G = (V, E)$ where $V$ is the set of vertices and $E$ is the set of edges, each edge $e \in E$ has a weight $e_w$. For an undirected graph with $n$ vertices, the maximum number of edges is $\frac{n(n-1)}{2}$. A dense graph has close to $\frac{n(n-1)}{2}$ edges. We can say a dense graph has $O(n^2)$ edges. In an undirected graph, each edge is bidirectional, meaning it connects two vertices in both directions.

The objective of the minimax path problem is to find a path $P$ from a starting node $i$ to a destination node $j$ such that the maximum weight of the edges in the path $P$ is minimized. A minimax path distance between a pair of points is the maximum weight in a minimax path between the points (Equation 2).

$$\Phi = \{max\_weight(p) \mid p \in \Theta_{(i,j,G)}\} \tag{1}$$

$$M(i, j \mid G) = min(\Phi) \tag{2}$$

where $G$ is the undirected dense graph. $\Theta_{(i,j,G)}$ is the set of all paths from node $i$ to node $j$. $p$ is a path from node $i$ to node $j$, $max\_weight(p)$ is the maximum weight in path $p$. $\Phi$ is the set of all maximum weights. $min(\Phi)$ is the minimum of Set $\Phi$ Liu (2023).

The distance can also be called the longest-leg path distance (LLPD) Little et al. (2020) or Min-Max-Jump distance (MMJ distance) Liu (2023). The all pairs minimax path distances calculate the distance between each pair of points in a dataset $X$ or graph $G$. It is also called all points path distance (APPD) Little et al. (2020). It is a matrix of shape $n \times n$. A dataset $X$ can be straightforwardly converted to a complete graph.

---

[1] https://en.wikipedia.org/wiki/Widest_path_problem

**Algorithm 4** MMJ distance by Calculation and Copy

**Input:** $\Omega$
**Output:** $\mathbb{M}_\Omega$

1: **function** MMJ_CALCULATION_AND_COPY($\Omega$)
2:    Initialize $\mathbb{M}_\Omega$ with zeros
3:    Construct a MST of $\Omega$, noted $T$
4:    Sort edges of $T$ from large to small, generate a list, noted $L$
5:    **for** e in $L$ **do**
6:        Remove $e$ from $T$. It will result in two connected subtrees, $T_1$ and $T_2$;
7:        For all pair of nodes $(p, q)$, where $p \in T_1$, $q \in T_2$. Fill in $\mathbb{M}_\Omega[p, q]$ and $\mathbb{M}_\Omega[q, p]$ with $e$.
8:    **end for**
9:    **return** $\mathbb{M}_\Omega$
10: **end function**

(a) Algorithm 4

```python
import networkx as nx

def cal_all_pairs_minimax_path_matrix_by_algo_4(distance_matrix):

    N = len(distance_matrix)
    all_pairs_minimax_matrix = np.zeros((N,N))

    MST = construct_MST_from_graph(distance_matrix)

    MST_edge_list = list(MST.edges(data='weight'))

    edge_node_list = [(edge[0],edge[1]) for edge in MST_edge_list]
    edge_weight_list = [edge[2] for edge in MST_edge_list]

    edge_large_to_small_arg = np.argsort(edge_weight_list)[::-1]

    edge_weight_large_to_small = np.sort(edge_weight_list)[::-1]
    edge_nodes_large_to_small = [edge_node_list[i] for i in edge_large_to_small_arg]

    for i, edge_nodes in enumerate(edge_nodes_large_to_small):
        edge_weight = edge_weight_large_to_small[i]
        MST.remove_edge(*edge_nodes)

        tree1_nodes = list(nx.dfs_preorder_nodes(MST, source=edge_nodes[0]))
        tree2_nodes = list(nx.dfs_preorder_nodes(MST, source=edge_nodes[1]))

        for p1 in tree1_nodes:
            for p2 in tree2_nodes:
                all_pairs_minimax_matrix[p1, p2] = edge_weight
                all_pairs_minimax_matrix[p2, p1] = edge_weight

    return all_pairs_minimax_matrix
```

(b) Python implementation of Algorithm 4

Figure 1: Algorithm 4 and its Python implementation. The three embedded for-loops make it look like an $O(n^3)$ algorithm, but it is actually an $O(n^2)$ algorithm.

```python
# G is an undirected dense graph, which has N vertices.
# adj_matrix is its adjacency_matrix.

def variant_of_Floyd_Warshall(adj_matrix):
    p = adj_matrix.copy()
    N = len(adj_matrix)

    for i in range(N):
        for j in range(N):
            if i != j:
                for k in range(N):
                    if i != k and j != k:
                        p[j,k] = min (p[j,k], max (p[j,i], p[i,k]))
    return p
```

Figure 2: A variant of the Floyd-Warshall algorithm for solving the minimax path problem

We can use a modified version of the Floyd–Warshall algorithm to solve the APPD in both directed and undirected dense graphs Weisstein (2008), or use the Algorithm 1 (MMJ distance by recursion) in Liu (2023), both of them take $O(n^3)$ time. However, in an undirected dense graph, we have a better choice. We may use an $O(n^2)$ algorithm to calculate the APPD matrix. There are several theoretical outcomes which claim the APPD matrix can be solved accurately in $O(n^2)$ Sibson (1973); Demaine et al. (2009; 2014); Alon and Schieber (2024). However, there is no code implementation of these algorithms, which implies they are impractical.

Code implementation is the process of translating a design or algorithm into a programming language. It is critical in algorithm design where ideas are turned into practical, executable code that performs specific tasks.

In section 4.3 (MMJ distance by calculation and copy) of Liu (2023), Liu proposes an algorithm which also claims to solve the APPD matrix accurately in $O(n^2)$, in an undirected dense graph. The algorithm is referred to as Algorithm 4 (MMJ distance by Calculation and Copy). In the paper, the algorithm is left unimplemented and untested. In this paper, we introduce a code implementation of Algorithm 4, and test it.

The widest path problem is a closely related topic to minimax path problem. In contrary, The objective of the widest path problem is to find a path $P$ from a starting node $s$ to a destination node $t$ such that the minimum weight of the edges in the path $P$ is maximized. Any algorithm for the widest path problem can be easily transformed into an algorithm for solving the minimax path problem, or vice versa, by reversing the sense of all the weight comparisons performed by the algorithm. Therefore, we can roughly say that the widest path problem and the minimax path problem are equivalent.

```python
# G is an undirected dense graph, which has N vertices.
import networkx as nx
def MST_shortest_path(G):

    MST = nx.minimum_spanning_tree(G)
    minimax_matrix = np.zeros((N, N))

    for i in range(N):
        for j in range(N):
            if j > i:
                max_weight = -1
                path = nx.shortest_path(MST, source=i, target=j)
                for k in range(len(path)-1):
                    if( MST.edges[path[k],path[k+1]]['weight'] > max_weight):
                        max_weight = MST.edges[path[k],path[k+1]]['weight']
                minimax_matrix[i,j] = minimax_matrix[j,i] = max_weight

    return minimax_matrix
```

Figure 3: Python implementation of *MST_shortest_path*, see Table 1

## 2 RELATED WORK

Numerous distance measures have been proposed in the literature, including Euclidean distance, Manhattan Distance, Chebyshev Distance, Minkowski Distance, Hamming Distance, and cosine similarity. These measures are frequently used in algorithms like k-NN, UMAP, and HDBSCAN. Euclidean distance is the most commonly used metric, while cosine similarity is often employed to address Euclidean distance's issues in high-dimensional spaces. Although Euclidean distance is widely used and universal, it does not adapt to the geometry of the data, as it is data-independent. Consequently, various data-dependent metrics have been developed, such as diffusion distances Coifman and Lafon (2006); Coifman et al. (2005), which arise from diffusion processes within a dataset, and path-based distances Fischer and Buhmann (2003); Chang and Yeung (2008).

Minimax path distance has been used in various machine learning models, such as unsupervised clustering analysis Little et al. (2020); Fischer et al. (2001; 2003); Fischer and Buhmann (2003), and supervised classification Chehreghani (2017); Liu (2023). The distance typically performs well with non-convex and highly elongated clusters, even when noise is present Little et al. (2020).

### 2.1 CALCULATION OF MINIMAX PATH DISTANCE

The challenge of computing the minimax path distance is known by several names in the literature, such as the maximum capacity path problem, the widest path problem, the bottleneck edge query problem Pollack (1960); Hu (1961); Camerini (1978); Gabow and Tarjan (1988), the longest-leg path distance (LLPD) Little et al. (2020), and the Min-Max-Jump distance (MMJ distance) Liu (2023).

A straightforward computation of minimax path distance is computationally expensive due to the large search space Little et al. (2020). However, for a fixed pair of points $x$ and $y$ connected in a graph $G = G(V, E)$, the distance can be calculated in $O(|E|)$ time Punnen (1991).

A well-known fact about minimax path distance is: "the path between any two nodes in a minimum spanning tree (MST) is a minimax path."Hu (1961) With this conclusion, we can simplify an undirected dense graph into a minimum spanning tree, when calculating the minimax path distance.

### 2.2 COMPUTING THE ALL POINTS PATH DISTANCE

Computing minimax path distance for all points is known as the all points path distance (APPD) problem. Applying the bottleneck spanning tree construction to each point results in an APPD runtime of $O(\min\{n^2 \log(n) + n|E|, n|E| \log(n)\})$ Little et al. (2020); Camerini (1978); Gabow and Tarjan (1988). The resulting APPD may not be accurate when calculating with bottleneck spanning tree, because a MST (minimum spanning tree) is necessarily a MBST (minimum bottleneck spanning tree), but a MBST is not necessarily a MST. A variant of the Floyd-Warshall algorithm

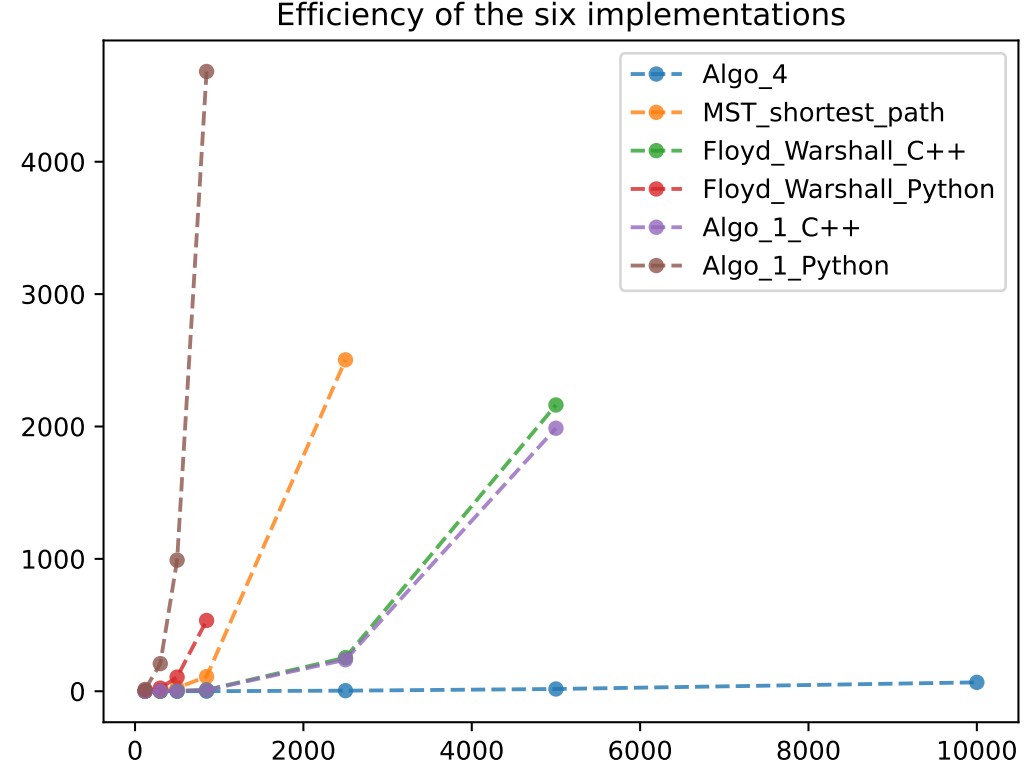

Figure 4: Performance of the algorithms (implementations)

| Implementation ID | Implementation name | Complexity | Coding language | Notes |
|---|---|---|---|---|
| 0 | Algo_1_Python | $O(n^3)$ | Python | Algorithm 1 (MMJ distance by recursion) |
| 1 | Algo_1_C++ | $O(n^3)$ | C++ | Algorithm 1 (MMJ distance by recursion) |
| 2 | Floyd_Warshall_Python | $O(n^3)$ | Python | A variant of Floyd-Warshall Algorithm |
| 3 | Floyd_Warshall_C++ | $O(n^3)$ | C++ | A variant of Floyd-Warshall Algorithm |
| 4 | MST_shortest_path | $O(n^3 log(n))$ | Python | Calculate the shortest path in a MST |
| 5 | Algo_4 | $O(n^2)$ | Python | Algorithm 4 (MMJ distance by Calculation and Copy ) |

Table 1: Profiles of the four algorithms. Two of them are implemented with different programming languages, Python and C++

can calculate the APPD accurately in $O(n^3)$ Aho and Hopcroft (1974). Several theoretical results suggest that the APPD matrix can be accurately solved in $O(n^2)$ time Sibson (1973); Demaine et al. (2009; 2014); Alon and Schieber (2024). However, the absence of code implementations for these algorithms indicates their impracticality.

| | data 139 (N = 120) | data 109 (N = 300) | data 18 (N = 500) | data 19 (N = 850) | data 16 (N = 2500) | data 35 (N = 5000) | data 136 (N = 10000) |
|---|---|---|---|---|---|---|---|
| Algo_1_Python | 13.451s | 208.363s | 990.308s | 4681.911s | >7200s | >7200s | >7200s |
| Algo_1_C++ | 0.033s | 0.414s | 1.794s | 9.032s | 237.961s | 1986.928s | >7200s |
| Floyd_Warshall_Python | 1.489s | 23.353s | 106.745s | 534.683s | >7200s | >7200s | >7200s |
| Floyd_Warshall_C++ | 0.033s | 0.436s | 2.324s | 10.035s | 253.909s | 2162.514s | >7200s |
| MST_shortest_path | 0.399s | 4.229s | 24.926s | 110.449s | 2503.483s | >7200s | >7200s |
| Algo_4 | 0.02s | 0.073s | 0.191s | 0.511s | 4.311s | 17.015s | 67.048s |

Table 2: Performance of the four algorithms. N is the number of points in the datasets.

## 3 IMPLEMENTATION OF THE ALGORITHM

As described in Section 1, the Algorithm 4 (MMJ distance by Calculation and Copy) in Liu (2023) also claims to solve the APPD matrix accurately in $O(n^2)$, in an undirected dense graph. But it is left unimplemented and untested. Figure 1a is Algorithm 4 (MMJ distance by Calculation and Copy) in Liu (2023), for convenience of reading, we re-post it here. Figure 1b is its python implementation.

Note the three embedded for-loops make it look like an $O(n^3)$ algorithm, but it is actually an $O(n^2)$ algorithm. Because when the variable $i$ in $Line$ 21 is small, both *tree1* and *tree2* are of size $O(n)$; but when the variable $i$ is large, both *tree1* and *tree2* are of size $O(1)$. The final net effect is that the three embedded for-loops only access each cell of the APPD matrix only once. Therefore, it is an $O(n^2)$ algorithm.

In the implementation, we first construct a minimum spanning tree (MST) of the undirected dense graph. The complexity of constructing a MST with prim's algorithm is $O(n^2)$. Then, we sort the edges of the MST in descending order. It is critical to remove the edges from the MST one-by-one, from large to small. Only by this we can get the two sub-trees, *tree1* and *tree2*. By traversing each sub-tree, nodes of the two sub-trees can be obtained, respectively.

## 4 TESTING OF THE ALGORITHM

In an experiment, we tested the Algorithm 4 (MMJ distance by Calculation and Copy) on seven datasets with different number of data points, note a dataset can be easily converted to a complete graph. The performance of Algorithm 4 is compared with three other algorithms that can calculate the APPD matrix.

Table 1 lists the profiles of the four algorithms. *Algo_1* is the Algorithm 1 (MMJ distance by recursion) in Liu (2023), it has complexity of $O(n^3)$; *Floyd_Warshall* is a variant of the Floyd-Warshall algorithm. Figure 2 is its python implementation. It has complexity of $O(n^3)$; *MST_shortest_path* firstly construct a minimum spanning tree (MST) of the undirected dense graph, then calculate the shortest path between each pair of nodes, then compute the maximum weight on the shortest path. Its complexity is $O(n^3 log(n))$. Figure 3 is its python implementation. The implementation is based on Madhav-99's code [2]; *Algo_4* is Algorithm 4 (MMJ distance by Calculation and Copy) in Liu (2023), it has complexity of $O(n^2)$. Both *Algo_1* and *Floyd_Warshall* are implemented with C++ and python, respectively, to test the difference between different programming languages.

### 4.1 PERFORMANCE

Table 2 is performance of the algorithms (implementations). We test each algorithm with seven datasets which have different number of data points. The data sources corresponding to the data IDs can be found at this URL (temporarily hidden for double blind review). The values are the time of calculating the minimax path APPD by each algorithm, on a desktop computer with "3.3 GHz Quad-Core Intel Core i5" CPU and 16 GB RAM.

To save time, we stop the execution of an algorithm if it cannot obtain the APPD matrix in 7200s (two hours). The computing time is recorded only once for each dataset and algorithm. Figure 4 converts the values in Table 2 into a figure. It can be seen that Algorithm 4 has achieved a good performance than other algorithms. It can calculate the APPD matrix of 10,000 points in about 67 seconds, while other algorithms cannot finish it in two hours.

Reasonably, the C++ implementations of *Algo_1* and *Floyd_Warshall* are much faster than their python edition. Interestingly, when implemented in python, *Algo_1* is much slower than *Floyd_Warshall*, but a little faster than *Floyd_Warshall* in C++.

### 4.2 SOLVING THE WIDEST PATH PROBLEM

As stated in Section 7 (Solving the widest path problem) of Liu (2023), Algorithm 4 (MMJ distance by Calculation and Copy) can be revised to solve the widest path problem APPD in undirected

---

[2] https://github.com/Madhav-99/Minimax-Distance

graphs, by constructing a maximum spanning tree and sort the edges in ascending order. In another experiment, we tested using Algorithm 4 to compute the widest path APPD. Result shows Algorithm 4 works good for solving the widest path problem.

## 5 DISCUSSION

### 5.1 MERIT OF ALGORITHM 1

Algorithm 1 (MMJ distance by recursion) has a merit of warm-start. Suppose we have calculated the APPD matrix $M_G$ of a large graph $G$, then we got a new point (or node) $p$, where $p \notin G$. The new graph is noted $G + p$. To calculate the APPD matrix of graph $G + p$, if we use other algorithms, we may need to start from zero. Algorithm 1 has the merit of utilizing the calculated $M_G$ for computing the new APPD matrix, with the conclusions of Theorem 3.3., 3.5., 6.1., and Corollary 3.4. in Liu (2023). This is especially useful when the graph is a directed dense graph, where starting from zero needs $O(n^3)$ complexity, but a warm-start of Algorithm 1 (MMJ distance by recursion) only needs $O(n^2)$ complexity. We can say Algorithm 1 supports online machine learning[3], in which data becomes available in a sequential order.

### 5.2 USING PARALLEL PROGRAMMING

If speed is the main concern of calculating the APPD matrix, we can use parallel programming to accelerate Algorithm 4. Firstly, we can use different processors for traversing the *tree1* and *tree2* in $Line\ 25\ and\ 26$ of Figure 1b. Secondly, we can copy the minimum spanning tree (MST) to many processors. For the $n$th processor, we just remove the $n$ largest edges, obtaining the $n$th *tree1* and *tree2*, traversing them, then fill in the corresponding positions of the APPD matrix that are decided by the $n$th *tree1* and *tree2*.

## 6 CONCLUSION

We implemented the Algorithm 4 (MMJ distance by Calculation and Copy) that was introduced by Liu in a previous paper. Then tested the implementation and compared it with several other algorithms that can calculate the all pairs minimax path distances, or also called the all points path distance (APPD). Experiment shows Algorithm 4 works good for solving the widest path or minimax path APPD matrix. As an algorithm of $O(n^2)$ complexity, it can drastically improve the efficiency of calculating the APPD matrix. Note algorithms for solving the APPD matrix are at least in $O(n^2)$ complexity, because the matrix is an $n \times n$ matrix.

In Section 2.3.3. of the paper "Path-Based Spectral Clustering: Guarantees, Robustness to Outliers, and Fast Algorithms," Little et al. (2020) Dr. Murphy and his collaborators write:

*"Naively applying the bottleneck spanning tree construction to each point gives an APPD runtime of $O(min\{n^2 log(n) + n|E|, n|E|log(n)\})$. However the APPD distance matrix can be computed in $O(n^2)$, for example with a modified SLINK algorithm (Sibson, 1973), or with Cartesian trees (Alon and Schieber, 1987; Demaine et al., 2009, 2014). "*

The author sent an email for further clarity about this statement.

The author:

*"You indicated the APPD distance matrix can be computed in $O(n^2)$. However, I searched the Internet and github, I have not found any code implementation that can accurately calculate the APPD distance matrix in $O(n^2)$. Do you know any code implementation of that? Please indicate it to me. "*

Dr. Murphy:

*"If you can find an implementation of SLINK to do single linkage clustering in $O(n^2)$, then you can do APPD by reading off the distances from the resulting dendrogram. I don't know any im-*

---

[3] https://en.wikipedia.org/wiki/Online_machine_learning

*plementations of SLINK, and it may be easier to prove things about than to implement practically.*
*"*

*"Regarding tree structures, these are certainly more of theoretical interest, and I would not be surprised if there were no practical implementations of them at all. So, achieving $O(n^2)$ via those methods may be impractical. "*

Therefore, we can roughly conclude: the Algorithm 4 (MMJ distance by Calculation and Copy) in Liu (2023), is the first algorithm that has actual code implementation to solve the APPD matrix of minimax path or widest path problem in $O(n^2)$, in an undirected dense graph.

It is also the fastest algorithm for solving the all points path distance (APPD) matrix by far.

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
