# OpenReview forum: "An efficient implementation for solving the all pairs minimax path problem in an undirected dense graph"
_ICLR.cc/2025/Conference — Submitted to ICLR 2025_

### Official Review · Reviewer_qiFx · 2024-10-30

**Soundness:** 1
**Presentation:** 2
**Contribution:** 1
**Rating:** 1
**Confidence:** 4

**Summary:**

Given a weighted graph and a vertex pair, the minimax path problem is to compute a path between the given vertex pair such that the maximum weight of the edges in the path is minimized. All pairs minimax path problem seeks to compute the maximum weight of the minimax path for every vertex pair. This paper implements a fast algorithm for the all pairs minimax path problem, which is proposed in the previous paper. Experiments show that their implementation is indeed faster.

**Strengths:**

Judging from the manuscript, this is the first implementation for $O(n^2)$ time algorithm for the all pairs minimax path problem. Because this algorithm works for any graph including dense graph (where $m=\Theta(n^2)$), this is actually an optimal algorithm.

**Weaknesses:**

The most prominent weakness of this work is that it only provides an implementation of the existing method of the previous paper of Liu (arXiv 2023). I think the implementation itself can be considered as a significant contribution when the original algorithm has some difficulties in implementing and they are tackled by some novel or dedicated techniques. However, the implemented algorithm (Figure 1(a)) is much simpler that seems to be easily implemented, and actually it is implemented by short Python codes (Figure 1(b)). We do not find any implementation techniques in Section 3; Section 3 consists of just an explanation for the existing method of Liu. This means that obtaining simple algorithm that is easy for implement has already been done in the previous work. Thus, I think this is not a significant contribution.

Another weakness of this work is that it is never compared empirically with the existing (and traditional) O(n^2) algorithm. In the conclusion, the authors claim that the SLINK algorithm of Sibson (1973) is difficult to implement by quoting the conversation with other people. However, we easily find some SLINK implementations such as https://github.com/battuzz/slink and https://github.com/jackyust/SLINK_CLINK . Moreover, the paper of Sibson also has a FORTRAN implementation of his algorithm. Thus, it is questionable that the SLINK algorithm is hard to implement. At least, the authors should try to implement SLINK and compare it with the proposed implementation.

Minor comments:
- The citations should be enclosed with brackets. Maybe you use \citet{...} instead of \citep{...}, don't you?
- Tables 1 and 2 are significantly wider than the paper width; please fit these tables into the paper width.

**Questions:**

Since the APPD matrix contains O(n^2) elements, explicitly computing APPD matrix in O(n^2) time is an optimal algorithm. Thus, I want to consider a variant of this problem: given a graph $G$ with $n$ vertices and $m$ edges, we preprocess $G$ to build a data structure $D$ that can answer the minimax path distance of any vertex pair in reasonable time. For relatively dense graphs, where $m=o(n^2)$ and $m=\omega(n)$, we can expect to lower the preprocessing time or space requirement below $n^2$ at the cost of query time of $D$. Do you know any existing works for this problem?

---

> ### Author Response · Authors · 2024-11-14
>
> Question:
>
> "Another weakness of this work is that it is never compared empirically with the existing (and traditional) O(n^2) algorithm."
>
> Response:
>
> Although Dr. Murphy indicated the SLINK algorithm can be revised to solve the APPD matrix in $O(n^2)$ time. I do NOT agree with him about this opinion. I do agree with him that there is no code implementation to solve the APPD matrix in $O(n^2)$ time before this paper. If you agree with Dr. Murphy that the SLINK algorithm can be revised to solve the APPD matrix in $O(n^2)$ time. Please provide your code so that we can test it.
>
> In simple words, there is no other existing (and traditional) O(n^2) code implementation to be compared with, the implementation introduced in this paper is the first and the only one by far.

---

### Official Review · Reviewer_Wbvm · 2024-10-31

**Soundness:** 3
**Presentation:** 2
**Contribution:** 1
**Rating:** 1
**Confidence:** 4

**Summary:**

This paper is about an implementation of an algorithm for all pairs minimax path problem in undirected dense graphs that works in $O(n^2)$ time. The problem asks to output the minimax distance between every pair of points s and t, where the minimax distance between s and t is the minimum of the maximum weighted edge in a path from s to t, over all the paths from s to t. The algorithm is from "Gangli Liu. 2023. Min-Max-Jump distance and its applications. arXiv preprint arXiv:2301.05994 (2023).", and essentially first finds an MST, and uses the fact that the minimax distance between any s and t is the maximum weighted edge on the st path in the MST. Then they use this fact to fill up an n by n matrix storing all these values. This paper basically codes this algorithm up.

**Strengths:**

the problem is well motivated in general.

**Weaknesses:**

while the problem is well motivated, finding an implementation for the algorithm is not very well motivated. It seems that no implementation existed before because the algorithms were not very efficient, and the algorithm of Liu is quite simple and efficient. The mere fact of coding that algorithm up is a simple classroom exercise it seems. If there are techniques used for going from the pseucode to the implementation, the authors have not described them, and it seems from the code that there are no such techniques.
Besides the above fundamental point, the write up could be polished, there are several grammatical mistakes.
abstract: "it is a code implementation of..." -> " this paper is a code implementation of the minimax path algorithm in Liu".
line 18:: works good -> works well.
line 106:: the sense -> the sign
in the related work it is good to note if there are any other papers like your paper where they implement an algorithm from a previous paper.

**Questions:**

I fundamentally don't see the point in this paper, it seems that all the heavy work and clever ideas were due to the Liu paper. Your correspondence in the conclusion section talks about a paper that was written before Liu, and I believe that implementing that would be hard, but it doesn't mean that implementing Liu's algorithm is hard. Implementing Liu's algorithm seems like a homework exercise, and please clarify any obstacles in doing this.

---

> ### Author Response · Authors · 2024-11-14
>
> The paper has two main contributions:
> 1. It provides the first code implementation for solving the all pairs minimax path problem or widest path problem in an undirected dense graph, in $O(n^2)$ time.
> 2. It provides the fastest code implementation for solving the all pairs minimax path problem or widest path problem in an undirected dense graph.
>
> Could a homework exercise make these two contributions? Please indicate the specific homework exercise which has made achievements like listed above.
>
> Liu's paper only proposed some ideas, it was unimplemented and untested. The implementation is not straightforward. You can test to use ChatGPT to implement the algorithm proposed in Liu's paper. It is not easy to implement it correctly.
>
> With the contribution of this paper, we have a usable tool to use it in practice.
>
> It is the State-Of-The-Art tool for solving the APPD matrix.

---

### Official Review · Reviewer_YNVS · 2024-11-03

**Soundness:** 1
**Presentation:** 1
**Contribution:** 1
**Rating:** 1
**Confidence:** 5

**Summary:**

Given an undirected graph, this paper considers the problem of computing the minimax path problem between all pairs of vertices. For any path, let the bottleneck edge be the largest edge on the path. For all pairs of vertices, compute a path with the smallest bottleneck edge. The authors refer to a prior paper that suggests a simple spanning tree based algorithm to compute this bottleneck edge and also state that this algorithm has an execution time of O(n^2). The main contribution of this algorithm is in the implementation of this algorithm.

**Strengths:**

The problem of finding all pairs of minimax path is interesting and has applications within ML

**Weaknesses:**

* A straight-forward implementation of an existing algorithm cannot be considered publishable research
* Writing quality and presentation is extremely poor
* Theoretical claims about complexity is not formally established.

**Questions:**

NA

---

### Official Review · Reviewer_ThN1 · 2024-11-04

**Soundness:** 1
**Presentation:** 1
**Contribution:** 1
**Rating:** 1
**Confidence:** 5

**Summary:**

The praper implements and experimentally tests an algorithm for computing minimax path distance that was proposed in paper Liu 2023. In my opinion the contribution of the paper is certianly below ICLR bar, as there are the following shortcomings:
- the paper is in area of exparimental algorithmis with no siginificant contribution to machine learning,
- there is no siginificant contribution to algorithms, i.e., the paper just implents in a straight-forward way the algorithms and tests it, there is not technincal challange in this implementation, nor the tests introduce any nowel methodology,
- the paper makes false statement that previous quadratic time algorithms have not been impmemented - this is not true as even the orginal paper by Sibson from 1973 contains the Fortran implementation of the algorithm.

**Strengths:**

The paper does not contain any significant contribution.

**Weaknesses:**

- the paper is in area of exparimental algorithmis with no siginificant contribution to machine learning,
- there is no siginificant contribution to algorithms, i.e., the paper just implents in a straight-forward way the algorithms and tests it, there is not technincal challange in this implementation, nor the tests introduce any nowel methodology,
- the paper makes false statement that previous quadratic time algorithms have not been impmemented - this is not true as even the orginal paper by Sibson from 1973 contains the Fortran implementation of the algorithm.

**Questions:**

No questions.

---

> ### Author Response · Authors · 2024-11-14
>
> Question:
>
> "the paper is in area of exparimental algorithmis with no siginificant contribution to machine learning"
>
> Response:
>
> This comment suggests reviewer ThN1 has very limited knowledge in the machine learning field.

---

> > ### Author Response · Authors · 2024-11-14
> >
> > And the awkward spelling suggests reviewer ThN1 has very limited knowledge in the English language too.

---

### Author Response · Authors · 2024-11-14

We will not thank the reviewers because none of the reviews is constructive or insightful.

The paper has two main contributions:
1. It provides the first code implementation for solving the all pairs minimax path problem or widest path problem in an undirected dense graph, in $O(n^2)$ time.
2. It provides the fastest code implementation for solving the all pairs minimax path problem or widest path problem in an undirected dense graph.

If the reviewers do NOT agree with these two claims, please provides the Python code to prove their opinions. If you do not have Python code, other code like java, Fortran, or MATLAB is OK, we have many experts in this community to translate it into Python, and compare it with the code implementation introduced in this paper.

Always remember Linus Torvalds' old saying: "Talk is cheap. Show me the code."

---

> ### Public Comment · ~Runsheng_Xu3 · 2024-11-25
>
> I don’t know what  Linus Torvalds would think if he saw the way the authors quoted him.

---

> ### Public Comment · ~Qiangeng_Xu1 · 2024-11-25
> **Refreshing Rebuttal!**
>
> I praise the author's sense of humor of commanding reviewers to provide code or they would be considered cheap. This made my day!
> It will be crazy if a reviewer really provide a snippet.

---

> > ### Author Response · Authors · 2024-11-26
> >
> > We are not commanding reviewers to provide code. When a person has some opinion, she/he needs to provide some evidence to prove her/his point. Reproducible experiments are the best way to prove something in science and technology. In CS, reproducible experiments can be conducted with source code. So, code is the evidence. People can choose not to provide the evidence, it is not mandatory. Note the "please" word in the comment, it is not "must."

---

> ### Public Comment · ~Zarif_Ikram1 · 2024-11-25
> **Authors are clearly underestimating their contributions**
>
> I think the authors are underestimating their contributions and wish to propose another.
>
> There’s been numerous literatures [1, 2] on writing good **respectable** rebuttals that help authors to write rebuttals while respecting the reviewers. However, there is a clear gap on how to _not_ write a rebuttal. I feel the authors do a commendable job of that--an important contribution of their research.
>
> I feel the third contribution makes this work stronger. There is an old saying, "Three is better than two."
>
> 1. https://deviparikh.medium.com/how-we-write-rebuttals-dc84742fece1
> 2. https://maartensap.com/notes/rebuttals.html

---

### Author Response · Authors · 2024-11-16
**The contribution of this paper is an achievement from 0 to 1.**

Before this paper, there are zero code implementation for solving the all pairs minimax path problem or widest path problem in an undirected dense graph, in $O(n^2)$  time.
After this paper, we have one such code implementation for solving the problem.

---

### Public Comment · ~Tomer_Galanti1 · 2024-11-26
**Relevance to the ICLR community?**

Hi Authors,

I’m curious about which category this contribution is intended to fall under. I’m not sure if it aligns well with the primary research topics covered by ICLR.

From their website:

> We consider a broad range of subject areas including feature learning, metric learning, compositional modeling, structured prediction, reinforcement learning, uncertainty quantification and issues regarding large-scale learning and non-convex optimization, as well as applications in vision, audio, speech, language, music, robotics, games, healthcare, biology, sustainability, economics, ethical considerations in ML, and others.

> A non-exhaustive list of relevant topics:

> unsupervised, self-supervised, semi-supervised, and supervised representation learning
transfer learning, meta learning, and lifelong learning
reinforcement learning
representation learning for computer vision, audio, language, and other modalities
metric learning, kernel learning, and sparse coding
probabilistic methods (Bayesian methods, variational inference, sampling, UQ, etc.)
generative models
causal reasoning
optimization
learning theory
learning on graphs and other geometries & topologies
societal considerations including fairness, safety, privacy
visualization or interpretation of learned representations
datasets and benchmarks
infrastructure, software libraries, hardware, etc.
neurosymbolic & hybrid AI systems (physics-informed, logic & formal reasoning, etc.)
applications to robotics, autonomy, planning
applications to neuroscience & cognitive science
applications to physical sciences (physics, chemistry, biology, etc.)
general machine learning (i.e., none of the above)

---

> ### Author Response · Authors · 2024-11-26
>
> Many unsupervised or supervised models in ML rely on the calculation of the APPD matrix. E.g.,
>
> unsupervised models:
> 1. Clustering with Neural Network and Index (CNNI) model, see https://arxiv.org/abs/2212.03853
> 2. MMJ-K-Means model, see https://arxiv.org/abs/2301.05994
>
> supervised models:
> 1. MMJ classifier, see https://arxiv.org/abs/2301.05994
> 2. The classifier introduced in Morteza Haghir Chehreghani's paper, "Classification with Minimax Distance Measures", see https://ojs.aaai.org/index.php/AAAI/article/view/10799.
>
> There are many other examples.
>
> So, the paper is in the scope of following list:
> 1. unsupervised
> 2. supervised
> 3. general machine learning

---

### Public Comment · ~Yesian_Rohn1 · 2024-11-26
**Innovative Responses to Reviewer Criticisms**

I didn't expect the author's only novelty to be his response to the reviewers. If I am targeted by a mean reviewer next time, I think I can learn from the author's way of responding.

---

> ### Public Comment · ~Jiale_Zhao1 · 2024-11-29
> **Innovative Responses to Reviewer Criticisms**
>
> True

---

> ### Public Comment · ~Ruihan_Wang1 · 2024-12-04
>
> Hahaha,  please stop making me laugh (≧▽≦)

---

> ### Public Comment · ~Hang_Zhang25 · 2025-06-27
>
> An efficient implementation for rebutting all comments with questions.

---

### Comment · Program_Chairs · 2024-11-26

The program committee has reviewed the discussion on this forum. We remind all participants to review and follow the code of conduct for the conference and professional standard. Further exchange breaching the norm of the professionalism will cause participants to be removed from the conference.

---

### Meta-Review · Area_Chair_XXdR · 2024-12-20

**Metareview:**

The paper studies an algorithm for computing minimax path distances in graphs that was proposed in prior work. The main contribution of the work is an implementation and experimental evaluation of the algorithm. There was strong consensus among the reviewers that this work does not meet the bar for acceptance. The paper provides a straightforward implementation of an existing algorithm and there is little relevance of this work to the ICLR community.

**Additional Comments On Reviewer Discussion:**

The reviewers raised significant concerns about the significance and relevance of this work to ICLR. After the discussion with the authors, it remained clear that this paper cannot be accepted.

---

### Decision · Program_Chairs · 2025-01-22

Reject